# Health workers' perception of malaria rapid diagnostic test and factors influencing compliance with test results in Ebonyi state, Nigeria

Izuchukwu Frank Obi[1,2,3]*, Kabiru Sabitu[2], Abdulhakeem Olorukooba[2], Ayo Stephen Adebowale[4], Rabi Usman[1], Ugochukwu Nwokoro[1,3], Olufemi Ajumobi[5,6], Suleiman Idris[2], Lawrence Nwankwo[7], IkeOluwapo O. Ajayi[1,4]

1 Nigeria Field Epidemiology and Laboratory Training Program, Abuja, FCT, Nigeria, 2 Department of Community Medicine, Ahmadu Bello University, Zaria, Nigeria, 3 Department of Community Medicine, University of Nigeria Teaching Hospital, Ituku-Ozalla, Enugu State, Nigeria, 4 Department of Epidemiology and Medical Statistics, Faculty of Public Health, University of Ibadan, Ibadan, Nigeria, 5 Malaria Consortium, Abuja, Nigeria, 6 National Malaria Elimination Program, Federal Ministry of Health, Abuja, Nigeria, 7 Malaria Elimination Program, Ministry of Health, Abakaliki, Ebonyi State, Nigeria

* obifrank03@gmail.com

**Data Availability Statement:** The study data have been uploaded to figshare and are accessible using

## Abstract

### Background

The standard practice in treating uncomplicated malaria is to prescribe artemisinin-based combination therapy (ACT) for only patients with positive test results. However, health workers (HWs) sometimes prescribe ACTs for patients with negative malaria rapid diagnostic test (mRDT) results. Available evidence on HWs perception of mRDT and their level of compliance with test results in Nigeria lacks adequate stratification by state and context. We assessed HWs perception of mRDT and factors influencing ACTs prescription to patients with negative mRDT results in Ebonyi state, Nigeria.

### Methods

A cross-sectional survey was conducted among 303 HWs who treat suspected malaria patients in 40 randomly selected public and private health facilities in Ebonyi state. Health workers' perception of mRDT was assessed with 18 equally weighted five-point likert scale questions with maximum obtainable total score of 90. Scores $\geq 72$ were graded as good and less, as poor perception. Data were analysed using descriptive statistics and logistic regression model at 5% significance level.

### Results

Mean age of respondents was 34.6±9.4 years, 229 (75.6%) were females, 180 (59.4%) community health workers and 67 (22.1%) medical doctors. Overall, 114 (37.6%) respondents across healthcare facility strata had poor perception of mRDT. Respondents who prescribed ACTs to patients with negative mRDT results within six months preceding the

the DOI: https://doi.org/10.6084/m9.figshare.9932954.v1.

**Funding:** The author(s) received no specific funding for this work.

**Competing interests:** The authors have declared that no competing interests exist.

survey were 154 (50.8%) [PHCs: 50 (42.4%), General hospitals: 18 (47.4%), tertiary facility: 51 (79.7%) and missionary hospitals: 35 (42.2%)]. Poor HWs' perception of mRDT promoted prescription of ACT to patients with negative mRDT results (AOR = 5.6, 95% C.I = 3.2–9.9). The likelihood of prescribing ACTs to patients with negative mRDT results was higher among HWs in public health facilities (AOR = 2.8, 95% C.I = 1.4–5.5) than those in the private.

## Conclusions

The poor perception of mRDT and especially common prescribing of ACTs to patients with negative mRDT results among HWs in Ebonyi state calls for context specific interventions to improve their perception and compliance with mRDT test results.

## Introduction

Globally, malaria is an important health threat with about 219 million cases and 435,000 associated deaths in the year 2017, approximately 92% of the cases and 93% of the deaths occurred in sub-Saharan Africa [1]. The remarkable decline in the global burden of malaria witnessed since the turn of the century appear to have stalled. The global fight against malaria has not only failed to make new gains since 2017, it has actually lost grounds in sub-Saharan Africa including Nigeria [1,2]. Most malaria-related deaths occur within 48 hours of onset of symptoms, hence the need for early diagnosis and prompt treatment of cases with effective antimalarial drugs [1,3]. It is currently recommended that every suspected malaria case be confirmed using light microscopy (gold standard) or Rapid Diagnostic Test (RDT) before treatment [4–7]. Operational complexities such as erratic electric power supply and dearth of expert malaria microscopists, have limited the use of light microscopy in malaria endemic countries like Nigeria. However, most of these challenges have been overcome by the introduction of mRDT leading to better targeting of malaria treatment [8–11]. Furthermore, quality controlled mRDTs used correctly have been shown to give results comparable or even better than light microscopy under routine conditions [12–16].

Despite the proven effectiveness of mRDTs in malaria diagnosis, health workers (HWs), especially in sub-Saharan Africa [17], sometimes prescribe antimalarial drugs for patients with negative mRDT results even when it is obvious that not giving such prescriptions will do no harm to the patient [17–19]. This inappropriate prescription could lead to unnecessary use of antimalarial drugs, economic wastage, increased risk of selection for drug resistant parasites and delayed detection of real cause of fever [17]. Studies have reported proportions of patients with negative mRDT results given antimalarial drug prescriptions in Nigeria and elsewhere to vary between 14% and 74% [17–21]. Although some studies have identified possible reasons why health workers prescribe antimalarial drugs for patients with negative mRDT results; most of these studies were either qualitative or simply elicited possible reasons for such prescriptions without evaluating the strength of influence of identified factors [17–19,22,23]. Some factors reported to influence HWs compliance with mRDT test results include HWs perception of mRDT, job cadre, work experience, training on malaria case management, knowledge of causes of fever, availability of algorithm to guide treatment decisions when test result is negative, patient expectation, and suspicion of treatment failure [11,17,18,20,23]. Hitherto these have not been explored in Ebonyi State, Nigeria.

Evidence has shown that the role of factors influencing performance of HWs vary with context and environment [24]. Furthermore, health workers' perception of mRDT [11,17,23] as well as their antimalarial prescription practices also vary with settings [20,25,26]. The 2018 WHO country-led 'High burden to High impact' approach, aimed at putting the global malaria control response back on track, in its pillar two emphasized use of strategic information to drive impact. This initiative while discouraging a one-size-fits-all approach to malaria control, encourages sub-national stratification of interventions based on evidence, to ensure well targeted response [2]. Ebonyi state has the highest prevalence of malaria in southeast Nigeria [27] and malaria is the most prevalent medical condition treated in healthcare facilities across the state [28]. Almost two decades after introduction of mRDT in malaria case management in Ebonyi state, no study was found to have explored health workers' perception of mRDT and their level of compliance with mRDT results in the state. Although earlier in 2013 and 2014, a nationwide cross-sectional survey provided some information on provider and patient perception of mRDT in Enugu, a neighbouring state in southeast Nigeria, evidence shows that Nigerian states have peculiarities and differ markedly in uptake of interventions [19,29,30]. This is evidenced in a recent study that examined the uptake of another malaria intervention in Nigeria, thus buttressing the need for state-specific information for evidence-based decision making [31].We therefore assessed health workers' perception of mRDT and factors influencing ACT prescription to patients with negative test results in Ebonyi state Nigeria; in order to provide state-specific support for improved management of malaria and non-malaria febrile illnesses.

## Materials and methods

### Study setting

This study was conducted in Ebonyi state, southeast Nigeria. Malaria is endemic in the state with perennial transmission. *Plasmodium falciparum* is the predominant parasite species [32]. The state has 13 local government areas (LGAs) including three urban and ten rural LGAs. The state has a projected population (2019) of 3,147,661 using a growth rate of 2.8% and 2006 population figure as the baseline [32,33]. There are 555 registered healthcare facilities in the state at the comprising two tertiary healthcare facilities, 13 General hospitals (one in each LGA), 424 primary healthcare centers, 6 missionary hospitals and 110 small private health facilities [32]. The six missionary hospitals, located in six different rural LGAs, have capacity for light microscopy and malaria RDT and they provide healthcare for over 60% of the population within their catchment areas [33]. The secondary health facilities have capacity for light microscopy and RDT-based malaria diagnosis, while the PHC facilities only perform RDT-based malaria diagnosis. In the tertiary facility, mRDT is used for malaria diagnosis at the children outpatient (CHOP), children emergency (CHER), and a model comprehensive healthcare center located at Izzi LGA which serves a rural outpost for the tertiary facility. The other units in the hospital use malaria microscopy for diagnosis [32]. There were 11,187 health workers in the state including proprietary and patent medicine vendors, role model caregivers and traditional birth attendants, but only 2099 health workers were licensed to treat patients [32]. The average number of HWs licensed to treat patients in the different strata of healthcare facilities in the state are as follows: four community health workers (CHWs) in each PHC, six (medical doctors and nurses) in each General hospital, about 30 (doctors, nurses and CHWs) in each missionary hospital and 80 (doctors, nurses and CHWs) in the departments of the tertiary facility where mRDT is used for malaria diagnosis. Ebonyi State Malaria Elimination Programme (SMEP), with the support development partners, procure, store and distribute commodities (including mRDT and ACTs) for free diagnosis and treatment of malaria in 266 (out

of the 439) public facilities and the six missionary hospitals in the state [32]. The SMEP train and supervise health workers in supported facilities and also conduct quarterly malaria diagnosis quality assurance in these facilities [32].

## Study design and population

A cross-sectional study was conducted among health workers whose job description includes diagnosis and treatment of febrile patients in selected private (missionary) and public healthcare facilities supported by SMEP to carry out mRDT-based malaria diagnosis free of charge to clients in Ebonyi State between February and April, 2017.

## Sample size determination

The Cochran formular[34] with design effect (DEFF) to account for the multistage cluster design was used to calculate the minimum sample size [35]:

$$n \ (minimum \ sample \ size) = DEFF * \frac{(Z_\alpha)^2 pq}{d^2}$$

$$DEFF \ (Design \ effect) = 1 + (m - 1) * ICC$$

m = average number of respondents per cluster (health facility; PHCs/ General hospital) = 4

ICC = Intraclass correlation coefficient = 0.042 [36]

$Z_\alpha$ = standard normal deviate at 95% confidence level = 1.96

p = 0.26 (proportion of health workers who reported prescribing ACTs for patients with negative mRDT results) [19]

q = compliment of p i.e. (1-p) = 0.74

d = level of precision set at 0.05

Applying the formular, n = [1+ (4–1) x 0.042] x [(1.96)$^2$x 0.26 x 0.74]/0.05$^2$

n = 1.126 x 295.65 = 332.90

The target population (health workers qualified to treat suspected malaria patients in public and private facilities in Ebonyi State) is finite (N = 2099).[32] Therefore applying finite population correction and adjusting for anticipated non-response of 5%, gave a required minimum sample size of 302.60 respondents. We therefore recruited 303 healthcare workers for the survey.

## Sampling technique

A three-stage sampling process (Fig 1) was used to select the study participants. In stage one (selection of LGAs); two LGAs were selected from each of the three senatorial districts in the state by simple random sampling (SRS) giving six LGAs. In stage two (selection of health facilities), healthcare facilities were first stratified according to facility types: PHCs, General hospital, tertiary facility and missionary hospital. We then selected five PHCs from each of the six selected LGAs by SRS. The only general hospital located in each of the six selected LGAs was included. Three missionary hospitals were randomly selected from a frame of six missionary hospitals in the state. The only teaching hospital in the state was also included giving a total of 40 health facilities (30 PHCs, 6 General hospitals, 3 missionary hospitals and 1 teaching hospital). At the facility level, all consenting eligible HWs were recruited for the PHCs, General hospitals and missionary hospital. For the tertiary facility however, to ensure representativeness, the allocated sample size was proportionately assigned to the three departments of the hospital

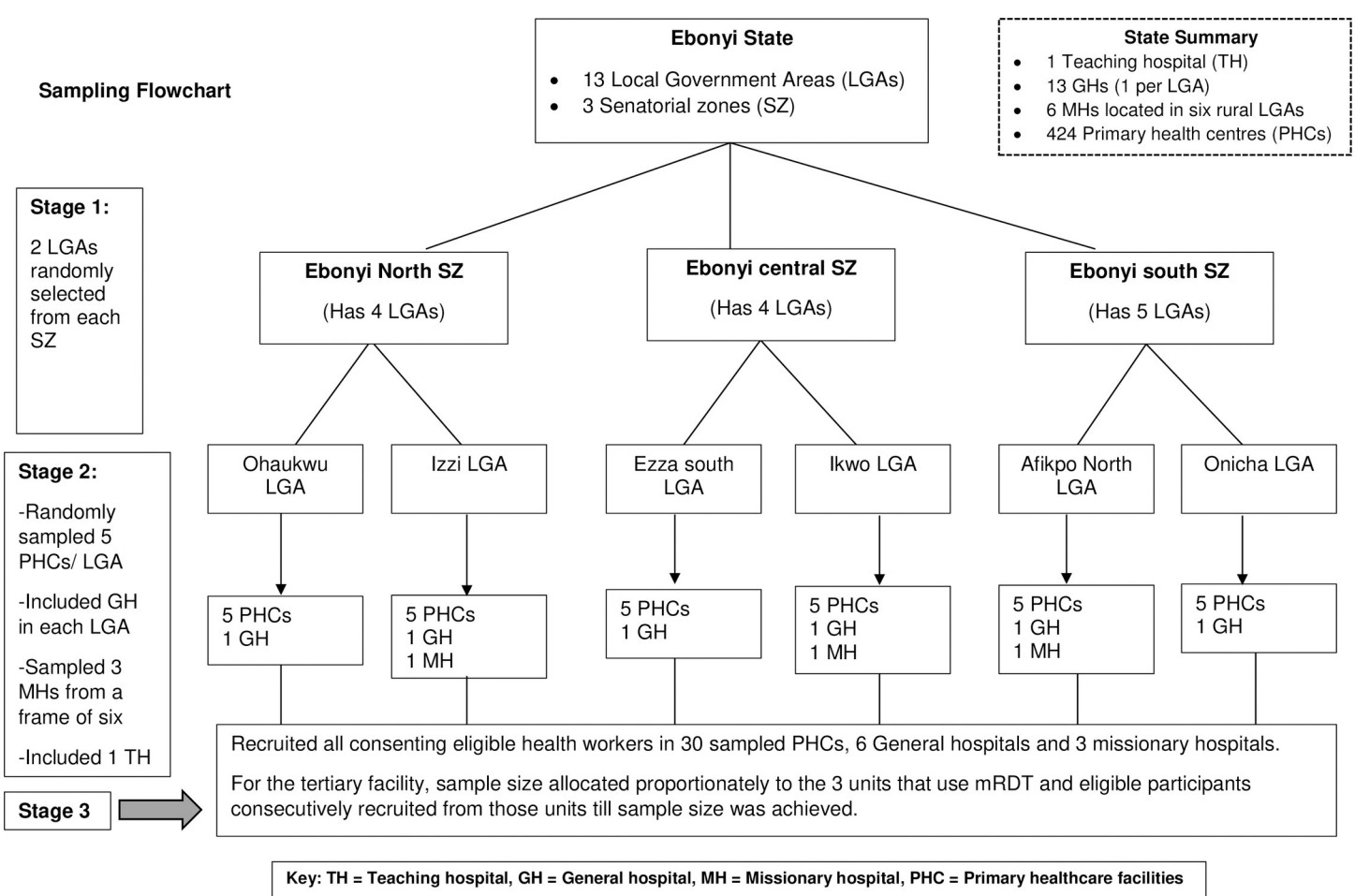

**Fig 1. Flowchart of the sampling process.**

where mRDT is used and eligible participants in each department recruited consecutively base on availability during study period until sample size was achieved.

## Data collection

The researcher and four trained research assistants (university graduates who had not worked at the state malaria program) administered an open data kit (ODK) based pre-tested semi-structured questionnaire on participants. The questionnaire was adapted from similar studies [11,18,19,21,23] and used to collect data on health workers' perception of mRDT, compliance with test results, training on malaria case management, availability of supportive supervision and job aids at work.

## Data processing and analysis

Data were entered into ODK collect on smart phone during interview. Skip logics, constraints and relevant-fields commands were used to limit wrong entries, incomplete entries and inconsistencies. Data were extracted from the mobile phones and converted to MS Excel (xls) format using ODK briefcase, cleaned and then exported to Epi Info version 7 for analysis. Descriptive statistics were summarized as frequency and percentages. Perception was assessed with 18

Likert scale questions with minimum and maximum obtainable total scores of 18 and 90 respectively. Computed scores of 72 and above (corresponding to agree or strongly agree for positively worded perception questions and vice versa) were graded as good perception while scores below 72 were graded as poor perception. Chi-squared test was used to determine association between ACT prescription to patients with negative RDT results and independent variables at 5% level of significance. Variables that were significant at $\alpha < 0.01$ during bivariate analysis were included in the logistic regression model using a step-wise approach. Results of the independent predictors of ACT prescription to patients with negative mRDT results were expressed using adjusted odds ratio and 95% confidence interval.

### Ethical considerations

Ethical approval (Ref: E7A/029/17/3) for the study was obtained from the Research Ethics Committees of Ebonyi State Ministry of Health and written permission obtained from the Ebonyi State Hospital Services Management Board. Written Informed consent was obtained

**Table 1. Sociodemographic characteristics of health workers, Ebonyi state (N = 303).**

| Characteristics | Frequency | Percentage |
|---|---|---|
| **Age groups in years** | | |
| < 30 | 97 | 32.0 |
| 30–39 | 114 | 37.6 |
| 40–49 | 75 | 24.7 |
| ≥ 50 | 17 | 5.7 |
| **Sex** | | |
| Females | 229 | 75.6 |
| Males | 74 | 24.4 |
| **Professional Cadre** | | |
| Community Health Workers* | 180 | 59.4 |
| Registered Nurses/ Midwives | 51 | 16.8 |
| Medical Doctors | 67 | 22.1 |
| Others** | 5 | 1.7 |
| **Highest Educational Qualification** | | |
| SSCE/ Certificates | 98 | 32.3 |
| Diploma | 113 | 37.3 |
| Graduate degree | 76 | 25.1 |
| Postgraduate Degree*** | 16 | 5.3 |
| **Years of Experience** | | |
| ≤ 10 | 180 | 59.4 |
| 11–20 | 90 | 29.7 |
| 21–30 | 28 | 9.2 |
| > 30 | 5 | 1.7 |
| **Distribution of respondents by facility type** | | |
| PHCs | 118 | 39.0 |
| General Hospitals | 38 | 12.5 |
| Tertiary facility | 64 | 21.1 |
| Missionary hospitals | 83 | 27.4 |

* Includes (Junior) CHEWS, Environmental Health Officers, Lab Technicians, Auxiliary Nurses, CHOs etc.

**Others = B. Ed, BSc and BMLS.

***Includes Postgraduate Medical College Fellowships and Memberships, MPH. MSc.

from each participant following explanations of the study aims, procedures, voluntariness, benefits and risks. Data privacy and confidentiality was maintained throughout the study.

## Results

### Sociodemographic characteristics of prescribers

All 303 respondents correctly completed the questionnaires giving a response rate of 100%. The mean age of respondents was 34.6 ± 9.4 years, the majority 229 (75.6%) were females, and 180 (59.4%) have practiced (as certified prescribers) for at most 10 years. Majority 180 (59.4%) were Community health workers, 51 (16.8%) were registered nurses/ mid-wives and 67 (22.1%) were medical doctors of different cadres. Thirty-nine percent of the respondents work in PHCs, 12.5% in General hospitals, 21.1% in tertiary facility and 27.4% in missionary hospitals (Table 1).

### Health workers' perception of Malaria-RDT

Table 2 shows health workers' perception of mRDT. While 277 (91.4%) of respondents agreed that febrile patients should always be tested before treatment, 58 (19.1%) see the use of clinical signs and symptoms alone as an accurate way of confirming malaria. Although 243 (80.2%) of respondents agreed that mRDT is an effective way of diagnosing malaria, 240 (79.2%) believed microscopy is more effective than mRDT. Furthermore, 262 (86.5%) of respondents said they fully trust a positive mRDT result as confirming malaria diagnosis whereas only 121 (39.9%) fully trusts a negative mRDT result as ruling out malaria diagnosis. Moreover, 216 (71.3%) of

**Table 2. Health workers' perception of malaria rapid diagnostic test, Ebonyi state, 2017.**

| Statements | *Agree n (%) | †Disagree n (%) |
|---|---|---|
| Clinical symptoms and signs accurate in confirming malaria diagnosis | 58 (19.1) | 245 (80.9) |
| **Always** confirm malaria diagnosis with lab test before treatment | 277 (91.4) | 26 (8.6) |
| mRDT effective in confirming malaria diagnosis | 243 (80.2) | 60 (19.8) |
| Microscopy is more effective than mRDT | 240 (79.2) | 63 (20.8) |
| Malaria-RDT is more effective than microscopy | 38 (12.5) | 265 (87.5) |
| Microscopy and mRDT are **equally** effective | 130 (42.9) | 173 (57.1) |
| I fully trust a **positive** mRDT result | 262 (86.5) | 41 (13.5) |
| I fully trust a **negative** mRDT result | 121 (39.9) | 182 (60.1) |
| More confident in positive mRDT result than in a negative one | 266 (87.8) | 37 (12.2) |
| More confident in a negative mRDT result than in a positive one | 17 (5.6) | 286 (94.4) |
| Equally confident in a positive and a negative mRDT result | 129 (42.6) | 174 (57.4) |
| No confidence in both positive and negative mRDT result | 5 (1.7) | 298 (98.3) |
| **Will prescribe antimalarial drugs to a patient with negative mRDT result if:** | | |
| I clinically suspect malaria | 216 (71.3) | 87 (28.7) |
| patient reported taking antimalarial drug before presentation to me | 141 (46.5) | 162 (53.5) |
| I suspect patient has low parasite level not detected by mRDT | 192 (63.4) | 111 (36.6) |
| patient pressurizes me to prescribe | 51 (16.8) | 252 (83.2) |
| I don't trust the mRDT I am using | 145 (47.9) | 158 (52.1) |
| I don't trust my skills in performing mRDT | 115 (38.0) | 188 (62.0) |

*Agree = Agree + strongly agree

†Disagree = Disagree + strongly disagree + Neutral

respondents said they will prescribe antimalarial drugs for patients with negative mRDT result if they have strong clinical suspicion of malaria while 192 (63.4%) said they will prescribe if they suspect patient has a low plasmodium parasite level not detected by the mRDT. Fifty-one (16.8%) of respondents said they will prescribe antimalarial drugs to patients who tested negative on mRDT if they are pressurized by patients to prescribe.

Overall, 114 (37.6%) respondents have poor perception of malaria rapid diagnostic test. This comprises 19 (16.1%) of prescribers in the PHCs, 12 (31.6%) of those in the General hospitals, 47 (73.4%) of those in the tertiary facility and 36 (43.4%) of those in the missionary hospitals (Fig 2).

## Prescription of ACTs for patients with negative RDT results

Overall, 154 (50.8%) of respondents reported prescribing ACTs to patients with negative RDT results within the six months preceding the interview. This comprised 50 (42.4%), 18 (47.4%), 51 (79.7%) and 35 (42.2%) of respondents from PHCs, General hospitals, tertiary facility and missionary hospitals respectively (Fig 2).

## Factors influencing ACT prescription for patients with negative mRDT results

At bivariate analysis (Table 3), the following variables were found to be significantly associated with ACT prescription to patients with negative RDT results: prescriber's professional cadre, perception of RDT, training on other causes of fever and health facility level (alpha = 5%). Variables not significantly associated with ACT prescription to patients with negative RDT results include prescriber's sex, professional experience, health facility type, availability of algorithm on what to do when RDT result is negative, training on malaria case management with RDT, duration of such training, training on negotiation skills and so on (alpha = 5%). However, at multivariate analysis (Table 3) only prescriber's perception of mRDT and health facility type were found to be independent predictors of ACT prescription to patients with negative mRDT result. Prescribers with poor perception of mRDT had 5.6 times the odds of prescribing ACTs to patients with negative RDT results compared to those with good perception of mRDT (AOR = 5.60, 95% C.I = 3.16–9.91). Also, prescribers in public facilities had 2.8 times the odds of prescribing ACT for patients with negative mRDT results compared to those in the private facilities (AOR = 2.81, 95% C.I = 1.44–5.46).

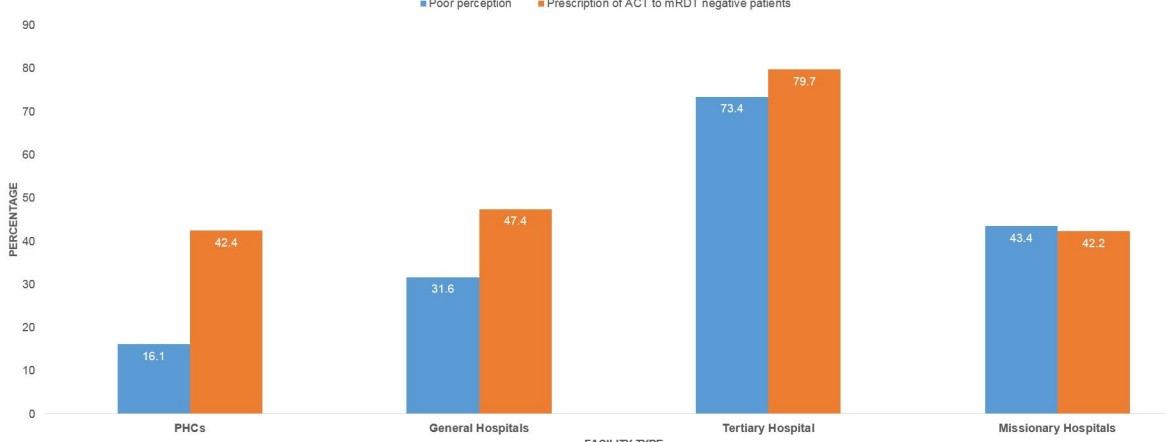

**Fig 2. Health workers' perception of mRDT and prescription of ACTs for patients with negative RDT results, Ebonyi state (N = 303).**

**Table 3. Factors influencing ACT prescription for patients with negative mRDT results, Ebonyi state.**

| Factors | Prescribe ACT when RDT result is negative | | Crude OR (95% C.I) | Adjusted OR (95% CI) |
|---|---|---|---|---|
| | Yes n (%) | No n (%) | | |
| **Type of health facility** | | | | |
| Public | 119 (77.3) | 101 (67.8) | 1.6 (0.8–2.7) *** | **2.7 (1.4–5.4)** ** |
| Private (*ref*) | 35 (22.7) | 48 (32.2) | 1 | |
| **Level of health facility** | | | | |
| Secondary/ Tertiary | 104 (67.5) | 81 (54.4) | 1.8 (1.1–2.8) ** | 1.7 (0.8–3.5) |
| Primary (*ref*) | 50 (32.5) | 68 (45.6) | 1 | |
| **Professional Cadre** | | | | |
| Medical Doctors, Nurses/ Midwives | 72 (46.7) | 51 (34.2) | 1.7 (1.1–2.7) ** | 1.3 (0.7–2.5) |
| Community Healthcare workers (*ref*) | 82 (53.3) | 98 (65.8) | 1 | |
| **Years of Experience** | | | | |
| ≤ 5 | 55 (35.7) | 52 (34.9) | 1.0 (0.7–1.7) | |
| > 5 (*ref*) | 99 (64.3) | 97 (65.1) | 1 | |
| **Sex** | | | | |
| Female | 110 (71.4) | 119 (79.9) | 0.6 (0.4–1.1) | |
| Male (*ref*) | 44 (28.6) | 30 (20.1) | 1 | |
| **Malaria case management training (CMT) with RDT** | | | | |
| Yes | 85 (55.2) | 85 (57.1) | 0.9 (0.6–1.5) | |
| No (*ref*) | 69 (44.8) | 64 (42.9) | 1 | |
| **Duration of malaria CMT with RDT** | | | | |
| < 3 days | 123 (79.9) | 108 (72.5) | 1.5 (0.9–2.6) | |
| > 3 days (*ref*) | 31 (20.1) | 41 (27.5) | 1 | |
| **Training on other causes of fever** | | | | |
| Yes | 69 (44.8) | 48 (32.2) | 1.7 (1.1–2.7) ** | 1.3 (0.7–2.2) |
| No (*ref*) | 85 (55.2) | 101 (67.8) | 1 | |
| **Training on negotiation skills** | | | | |
| No | 87 (56.5) | 80 (53.7) | 1.1 (0.7–1.8) | |
| Yes (*ref*) | 67 (43.5) | 69 (46.3) | 1 | |
| **Perception of mRDT** | | | | |
| Poor | 87 (56.5) | 27 (18.1) | 5.9 (3.4–9.9) * | **5.9(3.4–10.5)** * |
| Good (*ref*) | 67 (43.5) | 122 (81.9) | 1 | |
| **Supportive Supervision available** | | | | |
| No (*ref*) | 79 (51.3) | 84 (56.4) | 1 | |
| Yes | 75 (48.7) | 65 (43.6) | 0.8 (0.5–1.3) | |
| **Algorithm when RDT is negative** | | | | |
| No | 114 (74.0) | 99 (66.4) | 1.4 (0.9–2.4) | |
| Yes (*ref*) | 40 (26.0) | 50 (33.6) | 1 | |
| **Quality control checks available** | | | | |
| No | 36 (23.4) | 31 (20.8) | 1.2 (0.7–2.0) | |
| Yes (*ref*) | 118 (76.6) | 118 (79.2) | 1 | |
| **Feedbacks on mRDT Quality control** | | | | |
| No | 38 (24.7) | 38 (25.5) | 1.0 (0.6–1.6) | |
| Yes (*ref*) | 116 (75.3) | 111 (74.5) | 1 | |

*Significant at 0.1%;

**Significant at 5%;

***Significant at 10%;

## Discussion

The findings from this study showed that about one-third of the prescribers have poor perception of mRDT and this situation was worse in the tertiary facilities than other health facilities in Ebonyi state. Health worker's poor perception of mRDT was reflected in the level of reported wrong prescription practices found in this study. Similar responses have been reported in other studies [10,18]. Expectedly, health workers with poor perception of mRDT were likely to treat suspected malaria patients presumptively or even prescribe ACTs to patients with negative mRDT results. It is also evident in the current study that almost all the respondents agreed that it is important to confirm malaria diagnosis before treatment and a slightly lesser proportion agreed that mRDT is an effective way of confirming malaria diagnosis; it is then surprising that three-quarter of the respondents still said they will prescribe ACT for a patient with negative mRDT result if patient's signs and symptoms suggest malaria. This may be explained by a finding from this study that only few respondents fully trust a negative mRDT result as truly ruling out malaria, hence the temptation to go ahead with ACT prescription.

This study further found that four out of every five respondents expressed more confidence in a positive mRDT result than in a negative one. This is expected as a positive mRDT result in a febrile patient supports the clinical suspicion that necessitated the test. Similar finding was reported by a study assessing the effect of RDT results on health workers antimalarial prescription practices in selected PHCs in Uganda [10]. It is very important to address this seeming lack of trust in negative mRDT result by health workers as this is key to reducing inappropriate ACT prescription for patients with negative mRDT results. Interestingly, 13.5% of health workers expressed distrust for even a positive mRDT result in a febrile patient. A plausible explanation for this finding is the inherent poor perception of mRDT expressed by some health workers especially doctors in tertiary facilities. An ingrained negative bias towards mRDT results could make such prescribers indifferent to any mRDT result even when it supports their clinical suspicion. Another common unfavourable perception expressed by many prescribers in the state is that microscopy is more effective than mRDT in the diagnosis of malaria. This finding is different from the report from a study conducted among community health workers shortly after mRDT introduction in a neighbouring state where the majority of the interviewed prescribers perceived mRDT to be more effective than microscopy and clinical diagnosis [37]. The enthusiasm following initial introduction of mRDT may explain the favourable perception in the study above. Furthermore, our study included health workers from secondary and tertiary healthcare facilities who may likely perceive mRDT differently from lower level workers in PHCs.

This study also found prescribers' perception of mRDT to be a strong predictor of ACT prescription to patients with negative mRDT results. Prescribers with poor perception of mRDT were strikingly more likely to prescribe ACTs to patients with negative mRDT result compared to those with good perception. This further buttresses the fact that poor perception of mRDT is an important root cause that must be addressed if health workers are to stop prescribing ACT to patients with negative mRDT results. Various studies have reported health worker perception of mRDT to be a predictor of ACT prescription to patients with negative mRDT results [17,18,23]. Furthermore, facility type was also found to be a predictor of inappropriate ACT prescription to patients with negative mRDT results. Prescribers in private (missionary) hospitals were found to be less likely to prescribe ACTs to patients with negative mRDT results compared to those in public facilities. However, since ACT prescription was an elicited response, it is possible that prescribers in the private hospitals were a bit more conservative with the truth regarding their actual prescription practices.

## Limitations

Due to the cross-sectional nature of this study, some respondents might be tempted to report desired rather than actual practice of ACT prescription to patients with negative mRDT results. However, in order to minimize this source of bias, the respondents were re-assured that their responses will remain confidential and anonymous as no identifier information was collected.

## Conclusions

Poor perception of mRDT was common among health workers in Ebonyi state and was a determinant of healthcare workers' prescription of ACTs for patients with negative mRDT results. Health workers who practise in public facilities especially tertiary facility were more likely to prescribe ACTs to patients with negative mRDT results. We recommend deployment of facility-type tailored interventions aimed at improving HW perception of mRDT and compliance with negative test results.

## Supporting information

**S1 Questionnaire. Study questionnaire with informed consent page.**
(DOCX)

## Acknowledgments

The authors are grateful to Ebonyi State Hospital Services Management Board, State Malaria Elimination Programme, Malaria focal persons of selected LGAs, and the management and staff of selected health facilities for their immense support and contribution towards the success of this study. The findings of this study was presented and feedback received at the 7th Multilateral Initiative on Malaria Pan-African Conference which held at CICAD, Dakar, Senegal from 15–20 April, 2018. The authors acknowledge the support of the Nigerian Field Epidemiology and Laboratory Training Programme (NFELTP) towards training of the lead author and supporting writing of the manuscript. We are also grateful to the facilitators at the NFELTP manuscript writing workshop.

## Author Contributions

**Conceptualization:** Izuchukwu Frank Obi.

**Data curation:** Izuchukwu Frank Obi.

**Formal analysis:** Izuchukwu Frank Obi.

**Funding acquisition:** Izuchukwu Frank Obi.

**Investigation:** Izuchukwu Frank Obi, Olufemi Ajumobi, Lawrence Nwankwo.

**Methodology:** Izuchukwu Frank Obi, Kabiru Sabitu, Abdulhakeem Olorukooba, Ayo Stephen Adebowale, Olufemi Ajumobi, Suleiman Idris, IkeOluwapo O. Ajayi.

**Project administration:** Izuchukwu Frank Obi.

**Resources:** Izuchukwu Frank Obi.

**Software:** Izuchukwu Frank Obi, Abdulhakeem Olorukooba.

**Supervision:** Izuchukwu Frank Obi, Kabiru Sabitu, Suleiman Idris, Lawrence Nwankwo.

**Validation:** Izuchukwu Frank Obi.

**Visualization:** Izuchukwu Frank Obi, Ayo Stephen Adebowale.

**Writing – original draft:** Izuchukwu Frank Obi, Kabiru Sabitu.

**Writing – review & editing:** Izuchukwu Frank Obi, Kabiru Sabitu, Abdulhakeem Olorukooba, Ayo Stephen Adebowale, Rabi Usman, Ugochukwu Nwokoro, Olufemi Ajumobi, Suleiman Idris, Lawrence Nwankwo, IkeOluwapo O. Ajayi.

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
