## [Decision Letter · Decision Letter 0]

9 Jul 2019

PONE-D-19-15506

Health workers’ perception of malaria rapid diagnostic test and factors influencing compliance with test results in Ebonyi State, Nigeria

PLOS ONE

Dear Dr OBI,

Thank you for submitting your manuscript to PLoS ONE. After careful consideration, we felt that your manuscript requires substantial revision, following which it can possibly be reconsidered, thus governing the decision of a “major revision”. As point-out by the reviewer #2, the research question has been addressed multiple times in malaria RDT literature. Major concerns were related to methodological weaknesses in both the synthesis of the literature and the methods. Consequently, there are fundamental topics that need to be clarified and adjusted in the MS.  We therefore ask that you revise your manuscript paying close attention to the specific points raised by the reviewer #2.  For your guidance, a copy of the reviewers' comments was included below.

We would appreciate receiving your revised manuscript by July 30. To enhance the reproducibility of your results, we recommend that if applicable you deposit your laboratory protocols in protocols.io, where a protocol can be assigned its own identifier (DOI) such that it can be cited independently in the future. For instructions see: http://journals.plos.org/plosone/s/submission-guidelines#loc-laboratory-protocols

We look forward to receiving your revised manuscript.

Kind regards,

Luzia Helena Carvalho, Ph.D.

Academic Editor

PLOS ONE

**Journal Requirements:**

**Comments to the Author**

1. Is the manuscript technically sound, and do the data support the conclusions?

Reviewer #1: Yes

Reviewer #2: Partly

2. Has the statistical analysis been performed appropriately and rigorously? 

Reviewer #1: Yes

Reviewer #2: No

3. Have the authors made all data underlying the findings in their manuscript fully available?

Reviewer #1: Yes

Reviewer #2: Yes

4. Is the manuscript presented in an intelligible fashion and written in standard English?

Reviewer #1: Yes

Reviewer #2: Yes

5. Review Comments to the Author

Reviewer #1: The paper is well written and has addressed an important element affecting the use of RDTs in their area of practice.

Reviewer #2: The authors present the findings of a study that aims to assess the perceptions' of health workers towards malaria rapid diagnostic tests (mRDTs). In addition it also seeks to explore the characteristics that might influence compliance with test results This is an important public health paper that is generally well-written with a clear research question and reasoned structure to the paper. However, this research question has been addressed multiple times in the mRDT literature and it is unclear what the original research findings of this article are. THere are also methodological weaknesses in both the synthesis of the literature and the methods. The major and minor comments are outlined below which should be addressed before publication.

Major comments:

Introduction:

1) It is unclear from the introduction what this study adds to the existing literature on mRDT compliance. The references 14, 22-26 already indicate substantial evidence related to mRDTs and it makes one question what additional value the stratification of these results add to the existing literature. I would encourage the authors to provide a more critical appraisal of the existing literature, its' strengths and weaknesses and present more clear the current knowledge gaps.

Methods:

1) Page 7: The sample size calculation is unclear, for example what was the finite population and design effect used?

2) How was clustering within healthcare facilities accounted for?

3) It has not been stated how the variables used for adjustment were selected a priori were they all at the 5% level of significance.?

Results:

1) It was also interesting to note that 13.5% indicated that they did not trust positive results. This finding is also significant and requires some reflection in the discussion.

6. PLOS authors have the option to publish the peer review history of their article (what does this mean?). If published, this will include your full peer review and any attached files.

Reviewer #1: No

Reviewer #2: Yes: SHAM LAL

---

## [Author Response · Author response to Decision Letter 0]

7 Sep 2019

Reviewers’ comments and authors’ responses

Section: Introduction (Justification for the study) 

Comments:

1) It is unclear from the introduction what this study adds to the existing literature on mRDT compliance. The references 14, 22-26 already indicate substantial evidence related to mRDTs and it makes one question what additional value the stratification of these results add to the existing literature. I would encourage the authors to provide a more critical appraisal of the existing literature, its' strengths and weaknesses and present more clear the current knowledge gaps. 

Authors’ response:

The authors have carried out a substantial revision and reorganisation of this section in view of reviewer’s observations. This section now brings out more clearly gaps in existing literature which are filled by our study. Although some studies have identified possible reasons why health workers prescribe antimalarial drugs for patients with negative mRDT results, most of these studies were either qualitative or simply elicited possible reasons for such prescriptions without evaluating the strength of influence of identified factors. 

Furthermore, the 2018 WHO country-led ‘High burden to High impact’ approach, aimed at putting the global malaria control response back on track, in its pillar two emphasized use of strategic information to drive impact. This initiative while discouraging a one-size-fits-all approach to malaria control, encourages subnational stratification of interventions based on evidence, to ensure well targeted response. Moreover, a subnational profiling analysis for uptake of another malaria intervention revealed regional differences as the main predictor of differences in uptake, buttressing the need for state-specific information to support evidence-based decision making. Although Ebonyi state has the highest prevalence of malaria in southeast Nigeria and malaria is the most prevalent medical condition treated in healthcare facilities across the state, no study was found to have explored health workers’ perception of mRDT and their level of compliance with test results almost two decades after mRDT introduction in the state. 

Our study identified health workers’ perception of mRDT and the type of facility (public or private) as the predictors of ACT prescription to patients with negative mRDT result in Ebonyi. We also stratified health workers perception of mRDT and their prescription practices according to facility types to facilitate targeted intervention aimed at improving HWs perception of mRDT and compliance with test results for more impact.

This section now reads: 

‘Studies have reported proportions of patients with negative mRDT results given antimalarial drug prescriptions in Nigeria and elsewhere to vary between 14% and 74% [17–21]. Although some studies have identified possible reasons why health workers prescribe antimalarial drugs for patients with negative mRDT results; most of these studies were either qualitative or simply elicited possible reasons for such prescriptions without evaluating the strength of influence of identified factors [17–19,22,23]. Some factors reported to influence HWs compliance with mRDT test results include HWs perception of mRDT, job cadre, work experience, training on malaria case management, knowledge of causes of fever, availability of algorithm to guide treatment decisions when test result is negative, patient expectation, and suspicion of treatment failure [11,17,18,20,23]. Hitherto these have not been explored in Ebonyi State, Nigeria. 

Evidence has shown that the role of factors influencing performance of HWs vary with context and environment [24]. Furthermore, health workers’ perception of mRDT [11,17,23] as well as their antimalarial prescription practices also vary with settings [20,25,26]. The 2018 WHO country-led ‘High burden to High impact’ approach, aimed at putting the global malaria control response back on track, in its pillar two emphasized use of strategic information to drive impact. This initiative while discouraging a one-size-fits-all approach to malaria control, encourages sub-national stratification of interventions based on evidence, to ensure well targeted response [2]. Ebonyi state has the highest prevalence of malaria in southeast Nigeria [27] and malaria is the most prevalent medical condition treated in healthcare facilities across the state [28]. Almost two decades after introduction of mRDT in malaria case management in Ebonyi state, no study was found to have explored health workers’ perception of mRDT and their level of compliance with mRDT results in the state. Although earlier in 2013 and 2014, a nationwide cross-sectional survey provided some information on provider and patient perception of mRDT in Enugu, a neighbouring state in southeast Nigeria, evidence shows that Nigerian states have peculiarities and differ markedly in uptake of interventions [19,29,30]. This is evidenced in a recent study that examined the uptake of another malaria intervention in Nigeria, thus buttressing the need for state-specific information for evidence-based decision making [31].We therefore assessed health workers’ perception of mRDT and factors influencing ACT prescription to patients with negative test results in Ebonyi state Nigeria; in order to provide state-specific support for improved management of malaria and non-malaria febrile illnesses.’

 Kindly refer to pages 4 - 6, lines 70 - 100 in the clean manuscript.

Section: Methods 

Comments: 

1) Page 7: The sample size calculation is unclear, for example what was the finite population and design effect used? 

Authors' response: 

More details on the sample size calculation has been provided to improve clarity. This section now clearly shows the finite target population (health workers qualified to treat suspected malaria patients in public and private facilities in Ebonyi State N= 2099) and how we arrived at the design effect used.

 Kindly refer to pages 8 and 9, lines 135 - 153 in the clean manuscript.

2) How was clustering within healthcare facilities accounted for? 

Authors' response:

More information has been provided here for more clarity. Clustering within healthcare facilities was accounted for by applying design effect (DEFF) calculated using the average cluster size (m) and referenced intraclass correlation coefficient (ICC). 

Formular used was (DEFF=1+(m-1)*ICC).

 Kindly refer to page 8, lines 138 - 142 in the manuscript.

3) It has not been stated how the variables used for adjustment were selected a priori were they all at the 5% level of significance? 

Authors' response: 

Variables that were significant at α <0.01 during bivariate analysis were included in the logistic regression model using a step-wise approach. Results of the independent predictors of ACT prescription to patients with negative mRDT results were expressed using adjusted odds ratio and 95% confidence interval. 

 Kindly refer to page 11, lines 188 - 191 in the manuscript.

Section: Results 

Comments:

1) It was also interesting to note that 13.5% indicated that they did not trust positive results. This finding is also significant and requires some reflection in the discussion. 

Authors' response: Thank you for this important observation. We have reflected this finding in the discussion as suggested. 

 Kindly refer to pages 18 and 19, lines 273 - 278 in the manuscript.

Academic editor’s comments and authors’ responses

Comments:

1. As point-out by the reviewer #2, the research question has been addressed multiple times in malaria RDT literature. 

Authors' response: The authors have carried out a substantial revision of the introduction section. This section now brings out more clearly gaps in existing literature which are filled by our study.

 Kindly refer to pages 4 - 6, lines 70 - 100 in the clean manuscript.

2. Major concerns were related to methodological weaknesses in both the synthesis of the literature and the methods. Consequently, there are fundamental topics that need to be clarified and adjusted in the MS. We therefore ask that you revise your manuscript paying close attention to the specific points raised by the reviewer #2. 

Authors' response: The concerns raised by the second reviewer regarding literature synthesis and methodology (sample size calculation, accounting for intra-cluster correlation and the level of statistical significance for adjustment) have been addressed by providing more information for better clarity in the relevant sections of the manuscript as referenced above.

 Kindly refer to pages 8 and 9, lines 135 - 153 in the clean manuscript.

3. Authors' response: There are no changes in our financial disclosure.

 A rebuttal letter that responds to each point raised by the academic editor and reviewer(s). This letter should be uploaded as separate file and labeled 'Response to Reviewers'.

 A marked-up copy of your manuscript that highlights changes made to the original version. This file should be uploaded as separate file and labeled 'Revised Manuscript with Track Changes'.

 An unmarked version of your revised paper without tracked changes. This file should be uploaded as separate file and labeled 'Manuscript'. 

Authors' response: We have included a rebuttal letter, revised manuscript with tracked changes and the clean manuscript in our submission as requested.

Journal requirements: comments and authors’ responses

Comments:

Authors' response: The manuscript has been revised to ensure it meets PLOS ONE’s style requirements using the templates provided.

Authors' response: There are no changes to our Data Availability statement.

---

## [Decision Letter · Decision Letter 1]

1 Oct 2019

Health workers’ perception of malaria rapid diagnostic test and factors influencing compliance with test results in Ebonyi State, Nigeria

PONE-D-19-15506R1

Dear Dr. OBI,

We are pleased to inform you that your manuscript has been judged scientifically suitable for publication and will be formally accepted for publication once it complies with all outstanding technical requirements.

Within one week, you will receive an e-mail containing information on the amendments required prior to publication. At that time, the reference 19 should be adjusted (Reference 19, first author should be Mokuolu OA and not OOA).  When all required modifications have been addressed, you will receive a formal acceptance letter and your manuscript will proceed to our production department and be scheduled for publication.When all required modifications have been addressed, you will receive a formal acceptance letter and your manuscript will proceed to our production department and be scheduled for publication.

With kind regards,

Luzia Helena Carvalho, Ph.D.

Academic Editor

PLOS ONE

Additional Editor Comments (optional):

Reviewers' comments:

Reviewer's Responses to Questions

**Comments to the Author**

1. If the authors have adequately addressed your comments raised in a previous round of review and you feel that this manuscript is now acceptable for publication, you may indicate that here to bypass the “Comments to the Author” section, enter your conflict of interest statement in the “Confidential to Editor” section, and submit your "Accept" recommendation.

Reviewer #1: All comments have been addressed

2. Is the manuscript technically sound, and do the data support the conclusions?

Reviewer #1: Yes

3. Has the statistical analysis been performed appropriately and rigorously? 

Reviewer #1: Yes

4. Have the authors made all data underlying the findings in their manuscript fully available?

Reviewer #1: Yes

5. Is the manuscript presented in an intelligible fashion and written in standard English?

Reviewer #1: Yes

6. Review Comments to the Author

Reviewer #1: The authors have addressed the comments of the Reviewers and manuscript is much improved. Specifically they have justified the need for the a subnational level experience to guide the understanding of RDT use. They have also provided clarity on the sampling procedure and have taken some deeper dive on the factors influencing HW prescription preferences.

7. PLOS authors have the option to publish the peer review history of their article (what does this mean?). If published, this will include your full peer review and any attached files.

Reviewer #1: No

---

## [Editor Report · Acceptance letter]

10 Oct 2019

PONE-D-19-15506R1 

Health workers’ perception of malaria rapid diagnostic test and factors influencing compliance with test results in Ebonyi State, Nigeria 

Dear Dr. Obi:

I am pleased to inform you that your manuscript has been deemed suitable for publication in PLOS ONE. Congratulations! Your manuscript is now with our production department. 

With kind regards,

on behalf of

Dr. Luzia Helena Carvalho 

Academic Editor

PLOS ONE